# Potential Influence of Shading in Freshwater Ponds on the Water Quality Parameters and the Hematological and Biochemical Profiles of Nile tilapia (*Oreochromis niloticus* Linnaeus, 1758)

Geraldine B. Dayrit [1,2,*], Emmanuel M. Vera Cruz [3], Channarong Rodkhum [4], Mahmoud Mabrok [4,5], Pattareeya Ponza [6] and Mudjekeewis D. Santos [1,7]

1   The Graduate School, University of Santo Tomas, Manila 1008, Philippines
2   Department of Medical Microbiology, College of Public Health, University of the Philippines Manila, Ermita, Manila 1000, Philippines
3   Freshwater Aquaculture Center, Central Luzon State University, Nueva Ecija 3120, Philippines
4   Department of Veterinary Microbiology, Faculty of Veterinary Science, Chulalongkorn University, Bangkok 10330, Thailand
5   Department of Fish Diseases and Management, Faculty of Veterinary Medicine, Suez Canal University, Ismailia 41522, Egypt
6   Department of Agriculture Science, Faculty of Agriculture, Natural Resources and Environment, Naresuan University, Muang District, Phitsanulok 65000, Thailand
7   National Fisheries Research and Development Institute, Quezon City 1103, Philippines
*   Correspondence: gbdayrit@up.edu.ph

**Abstract:** Nile tilapia is a high-demand commodity in most developing countries including the Philippines, but its production is hampered by fluctuating water temperature (WT) and dissolved oxygen (DO) concentrations, which are mostly influenced by temperature warming. Net shading has been demonstrated to reduce pond temperature and increase the rate of spawning in cultured tilapia. The current study aimed to investigate the potential effects of shading on the water quality parameters, blood indices, and serum biochemical profiles of cultured Nile tilapia. Fish were collected from unshaded (control) and shaded (experimental) ponds in three breeding cycles. WT and DO concentrations were measured, while hematological and biochemical analyses were performed. The results showed that the average WT was considerably decreased in the shaded pond with no significant difference in the DO concentrations. Red blood cells, hemoglobin, and lymphocyte counts were significantly lower in fish raised in shaded ponds, although mean corpuscular volume and neutrophil counts were significantly higher. Likewise, Chole, TP, and Alb values were significantly affected by the interaction of cycle and shading setup. Cortisol, Glb, $K^+$, $Cl^-$, $Mg^{2+}$, and TCa values showed significant differences and were only affected by the cycle. Similarly, AST and ALT values showed significant differences and were affected by both cycle and shading setup. Water quality parameters (WT and DO), hematological blood indices, and serum biochemical variables were correlated positively with each other. In brief, prior data imply that net shading can reduce WT in aquaculture ponds, reduce stress in Nile tilapia, and lessen the consequences of temperature warming on species production.

**Keywords:** Nile tilapia; water quality; shading; hematology; biochemical analysis

**Key Contribution:** The present study gave insight into the potential use of net shading to reduce stress in Nile tilapia farms, and lessen the consequences of temperature warming on species production.

## 1. Introduction

The Philippines, like most of developing countries, is an agricultural country whose population relies heavily on local agricultural products that include locally grown crops and aquaculture products. Fish production plays a vital role in the country's nutritional

and economic profit, accounting for 11.5% of the average household food consumption per day at 392 g, generating PHP 214.9 million (USD 4.4 billion) as well as 1.1 million jobs in 2018 [1]. In the Philippines, tilapia is the most widely cultured freshwater fish, and the country's expanding population benefits from the valuable income and affordable supply of animal protein it provides [2]. Due to its high demand in the market, a decrease in tilapia production would have a major economic impact on the aquaculture industry locally [3,4] and globally as well [5,6].

All fish species exhibit maximum growth in an ideal range of temperature. Being an ectotherm, these fish depend on water temperature as one of the most important abiotic factors influencing their growth and survival, and any alterations in the temperature can cause certain effects in terms of physiological processes and behavioral activities [7].

Several studies have reported the adverse effects of temperature warming on aquaculture production due to climate change posing a major threat to fish growth and productivity, as well as the duration of sunshine playing an important role in the growth response and breeding cycle of tilapia [8]. Furthermore, variations in environmental conditions induce significant effects on the quantity, species composition, and nutrient composition [9]. Variations in WT are one of the most important factors influencing the aquaculture industry, and considerable changes in WT (3–5 °C) are potentially harmful and have a negative impact on aquatic fish growth and reproduction [10–14]. In this manner, abrupt changes in temperature have been shown to influence the maturation and breeding performances of tilapia breeders [15]. For example, in warmer environments, fish tend to have a longer growing season and have a faster rate, but they also seem to have a shorter lifespan than they do in cool water [16]. While the seasonal variation in water temperature allows fishes to adjust in terms of their metabolic capacities, their biochemical parameters (e.g., blood glucose, protein profile, aspartate aminotransferase (AST), alanine transaminase (ALT), triglycerides, and total cholesterol) are dependent on the existing environmental conditions.

Another equally important indicator of water quality is the concentration of dissolved oxygen (DO). The DO is also known to be influenced by WT, since oxygen is more soluble in cold water than in warm water [17]. As a result, DO concentrations are lower during the summer than in rainy months [18]. The negative correlation between the water temperature and oxygen content causes some fish to alter the functional activities of their organs and tissues to achieve a balanced state at elevated temperatures [19]. In the case of tilapia, its survival rate was significantly higher in high conditions than in low conditions, indicating a positive correlation between the DO content and its growth rate [20]. With this, adapting to changes in temperature and the DO content leads to alteration in its biochemical activities. Specifically, loss of appetite and impaired growth led to reduced serum glucose concentration and serum triglyceride, as they consume them for metabolism [21], and an increased serum protein concentration, which indicates damaged tissues in fish brought by high temperatures [22].

Changes outside of the preferred conditions of species can potentially cause stress to fish, resulting in reduced survival rates and low productivity [13,23]. Diseases serve as a microbiological gauge of climate change or global warming, as there is a significant correlation between the incidence of disease and ambient environmental temperatures [24]. Several factors influence disease's occurrence, and the plentitude of bacterial agents in the surrounding environment is often linked to water temperature [25]. Hematology is an indispensable diagnostic tool that aids in the study of a host's physiological responses to stress [26]. Moreover, with the abovementioned effects of the parameters on the biochemical composition of fishes, it can be deduced that biochemical profiles are useful physiologic markers that indicate the status of fish in a particular environmental condition [16,27–29].

Indeed, environmental stress affects the biological processes of fish and alters their homeostasis [30]. Evidence of alterations in hematological and biochemical processes has been recorded in fish after exposure to these environmental changes, and it has been reported that temperature has persistent effects on the heat acclimatization capacity of zebrafish during their embryonic stage [31]. Therefore, it is necessary to develop control

strategies based on a better understanding of the effects of environmental stressors on the health status of farmed fish [32]. Accordingly, this study aimed to set up a mitigation measure of structural shading over a tilapia pond using a greenhouse net to enhance fish welfare and ensure productive farming in aquaculture during environmental changes.

Net shading of tilapia ponds has been shown to reduce water temperature up to 4 °C (at noon) and increased the spawning rate and seed production of Nile tilapia compared to unshaded ponds [3,8]. However, there is a gap in the knowledge about the effect of shading on the physiological and immunological parameters of cultured fish. To the best of our knowledge, this study is one of the few that has briefly investigated the potential influence of shading on water quality parameters, hematological blood indices, and serum biochemical profiles of cultured Nile tilapia in the Philippines. This study also offers new perspectives on shading as a critical component of effective measures to combat the adverse effect of temperature warming.

## 2. Materials and Methods

### 2.1. Area of the Study

The study was conducted at the Freshwater Aquaculture Center (FAC) of Central Luzon State University (CLSU), Muñoz, Nueva Ecija (15.7326606° N, 120.9309769° E) (Figure 1), from May to August 2019 according to the national and institutional guidelines for the protection of animal welfare.

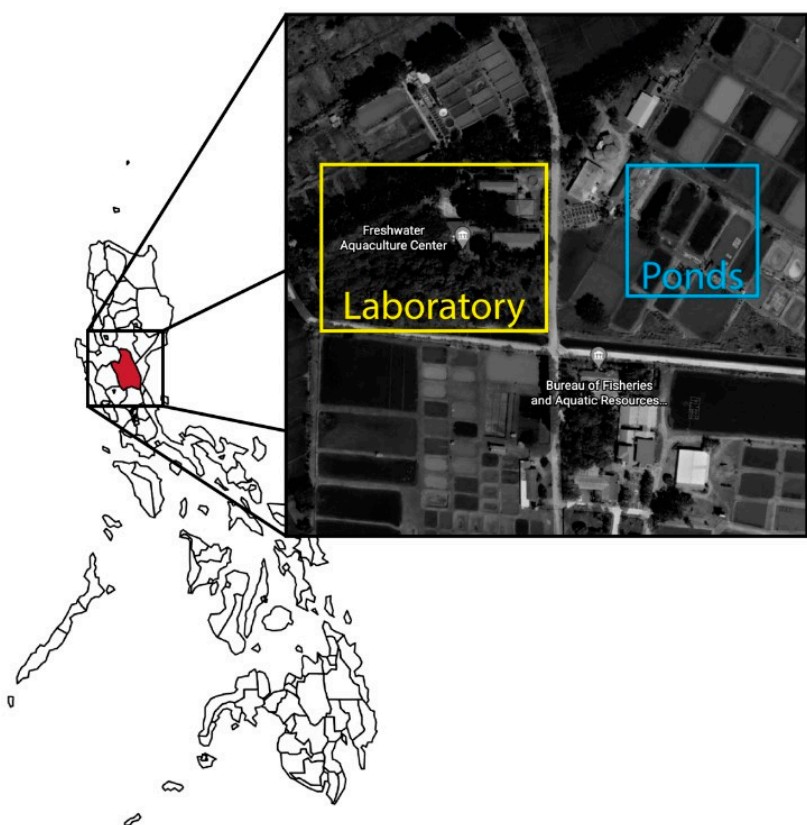

**Figure 1.** Map of the Philippines highlighting Nueva Ecija (red), location of ponds (blue), and laboratory (yellow) in Central Luzon State University.

### 2.2. Experimental Setup

A total of 144 apparent healthy Freshwater Aquaculture Center Selected Tilapia (FAST) strain of Nile tilapia breeders weighing 443.30 ± 70.81 g were equally distributed into two ponds (500 m$^2$; 1 m depth), each with three net enclosures or *hapas* (2 m × 3 m × 1 m) that served as the treatment units, having triplicates for the setup. The experimental pond was

shaded with a top greenhouse net, while the control pond was unshaded or had no top covering (Figure 2).

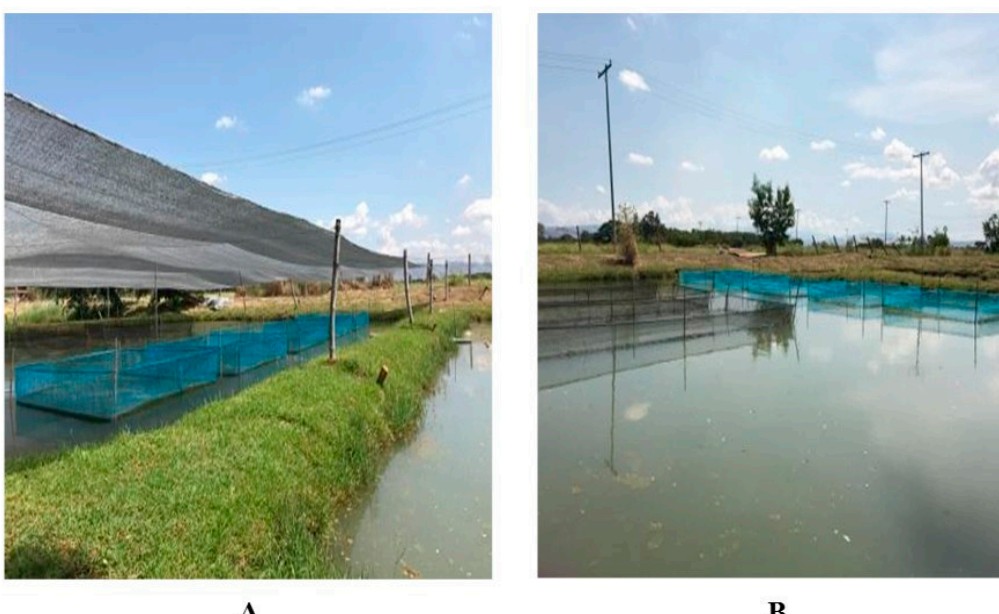

**Figure 2.** Experimental (**A**) and control (**B**) ponds in the Freshwater Aquaculture Center, Central Luzon State University, Muñoz, Nueva Ecija, the Philippines.

Each net enclosure was stocked with 24 breeders (6 males; 18 females). The fish were left to acclimatize in the continuous water flow system for two weeks in the same ponds before the experiment and fed at 2% of their body weight per day. The initial values of water temperature and dissolved oxygen of each pond as well as the sunshine duration collected from the Philippine Atmospheric, Geophysical and Astronomical Services Administration (PAGASA) of the Department of Science and Technology (DOST) are provided in Table 1. The experiment lasted for three successive cycles (63 days in total) as follows: Day 0–20 (Cycle 1), Day 21–42 (Cycle 2), and Day 43–63 (Cycle 3).

**Table 1.** Initial values of water temperature, dissolved oxygen, and sunshine duration among different ponds prior to the experimental study.

| Environmental Parameters | Environmental Condition | |
| --- | --- | --- |
| | Unshaded | Shaded |
| Water temperature (°C) | 32.89 | 30.73 |
| Dissolved oxygen (mg/L) | 4.26 | 4.99 |
| Sunshine duration (min) | 550 | |

### 2.3. Water Quality Analysis

Water temperature (WT) and dissolved oxygen (DO) concentration were measured twice daily (6 AM and 2 PM) for 63 days from each net enclosure using a digital thermometer and YSI oxygen meter, respectively.

### 2.4. Fish Biometric Index

Fish after being harvested were weighed to the nearest 0.01 g and were measured to the nearest millimeter for total length and width. The condition factor was calculated as follows:

$$\mathrm{Wt}_f \times 100\, \mathrm{L}_f^3, \tag{1}$$

where $\mathrm{Wt}_f$ is the weight of the fish, and $\mathrm{L}_f$ is the length.

### 2.5. Hematological Study

Prior to sampling at each time point (Days 0, 21, 42, and 63), fish were anesthetized using Tricaine Methane Sulfonate (MS-222), neutralized with sodium bicarbonate (Sigma Chemical Company, St. Louis, MO, USA), and then immediately measured and weighed. Subsequently, three fish per hapa (nine biological replicates and three technical replicates) were harvested for blood withdrawal. Blood samples were collected from the caudal vein using 23 G × 1 $\frac{1}{2}$ and a 5 mL syringe in tubes with EDTA (ratio 1.26 mg/0.6 mL) and immediately stored at 4–8 °C until used [8]. The following blood indices were measured: red blood cells (RBCs) count, hemoglobin (Hgb) concentration, hematocrit (Hct) percentage, mean corpuscular volume (MCV), mean corpuscular hemoglobin (MCH), mean corpuscular hemoglobin concentration (MCHC), and primary defending cells such as white blood cells (WBCs) count, neutrophils (Neutro), lymphocytes (Lympho), and platelets (Pct) concentrations. The analysis was performed following the protocol set by Mindray BC30s automated hematology analyzer (Mindray BC 30s Auto Hematology Analyzer Shenzhen Mindray Bio-Medical Electronics Co., Ltd., Shenzhen, China).

### 2.6. Biochemical Study

Blood samples were collected in plain and non-heparinized tubes and centrifuged at room temperature at $14,000\times g$ for 10 min. The serum samples were then transferred to new plain tubes and stored at $-20$ °C until analysis. Cortisol (Cor) was measured using a chemiluminescence immunoassay system (Maglumi 800 SNIBE Diagnostic, SNIBE Co., Ltd., Shenzhen, China). Glucose (Glu), cholesterol (Chole), total protein (TP), albumin (Alb), globulin (Glb), aspartate aminotransferase (AST), alanine aminotransferase (ALT), and alkaline phosphatase (ALP) were measured using a fully automated clinical chemistry analyzer (Selectra ProM ELITech Group, Puteaux, France). Levels of sodium ($Na^+$), potassium ($K^+$), chloride ($Cl^-$), total calcium (TCa), and magnesium ($Mg^{2+}$) were measured using an automated electrolyte analyzer (Statlyte C electrolyte analyzer, Kapitol Group Int. Ltd., Taoyuan, Taiwan).

### 2.7. Statistical Analyses

Values were expressed as means $\pm$ standard deviation (SD) to summarize all the gathered data from the experiment (Tables 2–7). The different parameters were compared between shading and non-shading using the *t*-test. The interaction of different parameters between shaded and unshaded setups was analyzed by two-way analysis of variance (ANOVA) using Strata version 14. Canonical correlation analyses were performed to investigate the relationships between variables. *p*-values of <0.05 were considered significant.

**Table 2.** Values of water temperature (WT) and dissolved oxygen (DO) of unshaded and shaded ponds across three different breeding cycles (mean $\pm$ SD).

| Water Quality | Cycle 1 | | Cycle 2 | | Cycle 3 | |
|---|---|---|---|---|---|---|
| Setups | Unshaded | Shaded | Unshaded | Shaded | Unshaded | Shaded |
| Water Temperature (°C) | 30.3 $\pm$ 0.55 [a] | 29.0 $\pm$ 0.33 [b] | 28.9 $\pm$ 0.66 [a] | 27.85 $\pm$ 0.49 [b] | 29.85 $\pm$ 0.60 [a] | 28.45 $\pm$ 0.38 [b] |
| Dissolved Oxygen (mg/L) | 1.69 $\pm$ 0.33 [a] | 2.45 $\pm$ 0.55 [b] | 1.39 $\pm$ 0.038 | 1.38 $\pm$ 0.21 | 2.27 $\pm$ 0.93 | 2.31 $\pm$ 1.11 |

*n* = 36; means with different scripts indicate significant difference at *p* < 0.05.

**Table 3.** Overall mean values of water temperature (WT) and dissolved oxygen (DO) of unshaded and shaded ponds (mean $\pm$ SD).

| Water Quality | Unshaded | Shaded |
|---|---|---|
| Water Temperature (°C) | 29.68 $\pm$ 0.83 [a] | 28.43 $\pm$ 0.62 [b] |
| Dissolved Oxygen (mg/L) | 1.78 $\pm$ 0.65 [a] | 2.04 $\pm$ 0.81 [a] |

*n* = 36; means with different scripts indicate significant difference at *p* < 0.05.

**Table 4.** Interaction of shading setup and breeding cycles to hematological parameters of Nile tilapia (*Oreochromis niloticus* Linnaeus, 1758) mean (±SD).

| S | C | WBCs (10$^9$/L) | RBCs (10$^{12}$/L) | Hgb (g/L) | Hct (%) | Pct (10$^9$/L) | MCV (fL) | MCH (pg) | MCHC (g/L) | Neutro (10$^9$/L) | Lympho (10$^9$/L) |
|---|---|---|---|---|---|---|---|---|---|---|---|
| No | 1 | 64.08 ± 10.64 [a] | 1.48 ± 0.43 [a] | 64.17 ± 17.94 [a] | 26.43 ± 5.14 [a] | 7.67 ± 3.61 [a] | 185.00 ± 28.49 [a] | 43.50 ± 1.55 [a] | 238.83 ± 29.14 [a] | 12.88 ± 4.33 [a] | 87.13 ± 4.33 [a] |
| Yes | 1 | 67.92 ± 12.34 [a] | 1.63 ± 0.25 [b] | 72 ± 10.02 [b] | 32.95 ± 7.35 [b] | 7.67 ± 2.80 [a] | 200.04 ± 22.04 [b] | 44.27 ± 2.12 [a] | 223.00 ± 28.74 [a] | 13.20 ± 6.07 [b] | 86.80 ± 6.07 [b] |
| No | 2 | 70.88 ± 18.75 [a] | 1.79 ± 0.19 [c] | 82.50 ± 11.57 [c] | 32.12 ± 2.84 [c] | 7.83 ± 2.93 [b] | 180.95 ± 20.76 [a] | 46.03 ± 2.89 [b] | 256.17 ± 22.94 [a] | 12.72 ± 3.55 [c] | 87.28 ± 3.55 [c] |
| Yes | 2 | 54.90 ± 16.69 [a] | 1.14 ± 0.09 [d] | 53.17 ± 3.54 [d] | 24.82 ± 3.43 [d] | 11.67 ± 5.54 [b] | 206.08 ± 29.22 [b] | 46.45 ± 1.34 [b] | 228.17 ± 36.49 [a] | 25.05 ± 7.88 [d] | 74.95 ± 7.88 [d] |
| No | 3 | 75.58 ± 7.29 [a] | 2.17 ± 0.18 [e] | 107.17 ± 11.11 [e] | 39.27 ± 5.59 [e] | 4.50 ± 1.38 [c] | 175.38 ± 12.94 [a] | 48.18 ± 2.29 [c] | 235.73 ± 110.98 [a] | 12.27 ± 2.22 [e] | 87.73 ± 2.22 [e] |
| Yes | 3 | 64.55 ± 8.13 [a] | 1.83 ± 0.08 [f] | 88.50 ± 1.87 [f] | 32.72 ± 1.68 [f] | 7.33 ± 4.03 [c] | 179.02 ± 3.62 [b] | 48.53 ± 1.75 [c] | 271.50 ± 13.31 [a] | 13.37 ± 7.07 [f] | 86.63 ± 7.07 [f] |

*n* = 3; means with different scripts indicate significant difference at *p* < 0.05. S, shaded; C, cycle.

**Table 5.** Overall, mean values (±SD) of hematological parameters of Nile tilapia (*Oreochromis niloticus* Linnaeus, 1758) between shading setups.

| Parameters | Pond Setup | |
|---|---|---|
| | **Unshaded** | **Shaded** |
| WBCs (10$^9$/L) | 70.18 ± 13.26 | 62.46 ± 13.36 |
| RBCs (10$^{12}$/L) | 1.82 ± 0.40 [a] | 1.53 ± 0.33 [b] |
| Hgb (g/L) | 84.61 ± 22.34 [a] | 71.22 ± 15.97 [b] |
| Hct (%) | 32.61 ± 6.97 | 30.16 ± 5.94 |
| Pct (10$^9$/L) | 6.67 ± 2.64 | 8.89 ± 4.12 |
| MCV (fL) | 180.44 ± 20.76 [b] | 195.29 ± 23.30 [a] |
| MCH (pg) | 45.91 ± 2.93 | 46.42 ± 2.44 |
| MCHC (g/L) | 243.58 ± 64.13 | 240.89 ± 34.46 |
| Neutrophil (10$^9$/L) | 12.62 ± 3.28 [b] | 17.21 ± 8.74 [a] |
| Lymphocyte (10$^9$/L) | 87.38 ± 3.28 [a] | 82.79 ± 8.74 [b] |

*n* = 3; means with different scripts indicate significant difference at *p* ≤ 0.05.

**Table 6.** Interaction of shading setup and breeding cycles to biochemical profiles of Nile tilapia (*Oreochromis niloticus* Linnaeus, 1758) mean ± SD.

| S | C | Cor (ng/mL) | Glu (mmol/L) | Chole (mmol/L) | TP (g/L) | Alb (g/L) | Glb (U/L) | AST (U/L) | ALT (U/L) | ALP (U/L) | K$^+$ (mmol/L) | Na$^+$ (mmol/L) | Cl$^-$ (mmol/L) | Mg$^{2+}$ (mmol/L) | TCa (mmol/L) |
|---|---|---|---|---|---|---|---|---|---|---|---|---|---|---|---|
| No | 1 | 299.24 ± 4.4 [a] | 4.19 ± 1.15 [a] | 2.22 ± 0.58 [a] | 20.23 ± 3.44 [a] | 10.87 ± 1.79 [a] | 8.49 ± 3.40 [a] | 13.28 ± 1.45 [aa] | 6.33 ± 2.63 [aa] | 2.41 ± 16.98 [a] | 3.77 ± 0.62 [a] | 165.28 ± 8.09 [a] | 157.93 ± 7.42 [a] | 2.93 ± 0.58 [a] | 3.98 ± 0.70 [a] |
| Yes | 1 | 232.89 ± 49.60 [a] | 5.27 ± 2.23 [a] | 2.16 ± 0.71 [b] | 19.63 ± 2.22 [b] | 11.17 ± 1.76 [b] | 9.06 ± 1.50 [a] | 21.99 ± 7.68 [ba] | 9.88 ± 1.07 [ba] | 56.54 ± 11.53 [a] | 3.27 ± 1.26 [a] | 160.17 ± 7.20 [a] | 150.85 ± 7.53 [a] | 3.33 ± 0.55 [a] | 3.63 ± 0.48 [a] |
| No | 2 | 177.03 ± 95.00 [b] | 4.24 ± 1.09 [a] | 2.08 ± 0.93 [c] | 20.54 ± 3.36 [c] | 8.12 ± 3.20 [c] | 10.90 ± 4.39 [b] | 17.30 ± 6.84 [ab] | 13.39 ± 4.13 [ab] | 54.57 ± 11.08 [a] | 4.05 ± 0.59 [b] | 156.25 ± 5.61 [a] | 144.43 ± 7.22 [b] | 3.10 ± 1.06 [b] | 2.84 ± 0.40 [b] |
| Yes | 2 | 176.78 ± 78.94 [b] | 3.84 ± 0.53 [a] | 2.16 ± 1.07 [d] | 25.76 ± 4.48 [d] | 12.60 ± 2.30 [d] | 13.14 ± 2.61 [b] | 33.33 ± 9.20 [bb] | 22.06 ± 10.24 [b] | 58.44 ± 10.14 [a] | 4.06 ± 0.99 [b] | 157.28 ± 3.64 [a] | 145.47 ± 6.54 [b] | 3.90 ± 1.73 [b] | 2.79 ± 0.47 [b] |
| No | 3 | 245.17 ± 77.86 [c] | 4.09 ± 0.83 [a] | 4.31 ± 0.94 [e] | 36.20 ± 4.12 [e] | 14.93 ± 1.66 [e] | 21.28 ± 2.82 [c] | 6.88 ± 1.68 [ac] | 5.43 ± 2.24 [ac] | 67.39 ± 11.78 [a] | 2.91 ± 0.63 [c] | 163.88 ± 5.63 [a] | 148.85 ± 4.34 [c] | 5.42 ± 1.72 [c] | 4.50 ± 1.29 [c] |
| Yes | 3 | 308.60 ± 157.51 [c] | 4.02 ± 0.76 [a] | 2.34 ± 0.34 [f] | 30.63 ± 4.74 [f] | 12.87 ± 2.70 [f] | 17.76 ± 2.19 [c] | 15.56 ± 6.15 [bc] | 9.09 ± 3.31 [bc] | 63.82 ± 7.35 [a] | 2.59 ± 0.99 [c] | 157.25 ± 5.17 [a] | 145.03 ± 4.13 [c] | 5.70 ± 2.53 [c] | 4.28 ± 1.42 [c] |

*n* = 3; means with different scripts indicate significant difference at *p* < 0.05. S, shaded; C, cycle.

**Table 7.** Overall, mean values (±SD) of biochemical profiles of Nile tilapia (*Oreochromis niloticus* Linnaeus, 1758) between shading setups.

| Parameters | Pond Setup | |
|---|---|---|
| | **Unshaded** | **Shaded** |
| Cor (ng/mL) | 244.21 ± 84.07 | 239.42 ± 113.76 |
| Glu (mmol/L) | 4.17 ± 0.97 | 4.38 ± 1.47 |
| Chole (mmol/L) | 2.87 ± 1.31 | 2.22 ± 0.73 |
| TP (g/L) | 25.65 ± 8.41 | 25.34 ± 5.95 |
| Alb (g/L) | 11.31 ± 3.61 | 12.21 ± 2.28 |
| Glb (g/L) | 13.56 ± 6.63 | 13.32 ± 4.18 |
| AST (U/L) | 12.44 ± 6.06 [a] | 23.73 ± 10.66 [b] |
| ALT (U/L) | 8.50 ± 4.75 [a] | 13.68 ± 8.47 [b] |
| ALP (U/L) | 61.46 ± 13.83 | 59.60 ± 9.76 |
| $K^+$ (mmol/L) | 3.58 ± 0.77 | 3.31 ± 1.19 |
| $Na^+$ (mmol/L) | 161.81 ± 7.38 | 158.23 ± 5.38 |
| $Cl^-$ (mmol/L) | 150.41 ± 8.40 | 147.12 ± 6.46 |
| $Mg^{2+}$ (mg/dL) | 3.82 ± 1.63 | 4.31 ± 1.98 |
| TCa (mmol/L) | 3.77 ± 1.09 | 3.57 ± 1.06 |

*n* = 3; means with different scripts indicate significant difference at *p* < 0.05.

## 3. Results

### 3.1. Water Quality

There were significant (*p* < 0.05) differences in WT values between shaded and unshaded ponds across all breeding cycles, whereas in the case of DO concentrations, the differences were only recorded in Cycle 1 (Table 2).

The mean values of WT were significantly lower in the shaded pond (28.43 ± 0.62) than in the unshaded pond (29.68 ± 0.83) (Table 3). Although the overall mean values of DO were higher in the shaded ponds, significant difference was only noticed during Cycle 1.

### 3.2. Biometric Index

The condition factors were higher in the shaded ponds (1.81 ± 0.59) than in the unshaded ones (1.61 ± 0.28); however, no significant difference was observed in the different collection periods and shading setups.

### 3.3. Hematological Study

The hematological profile was calculated and tabulated in the control and treated ponds (Tables 4 and 5). RBCs, Hgb, neutrophils, and lymphocyte values increased significantly in the shaded ponds to *p* = 0.0275, *p* = 0.0462, *p* = 0.0448, and *p* = 0.0446, respectively, while WBCs and MCHC values were not affected when compared to the unshaded ponds. RBCs, Hgb, and Hct values were significantly higher (*p* < 0.05) in the shaded ponds during Cycle 1 but were statistically lower (*p* < 0.05) in Cycles 2 and 3. In contrast, neutrophil and lymphocyte counts showed a consistent pattern of increase and decrease, respectively, in shaded ponds during all breeding cycles.

Furthermore, Pct (*p* = 0.472) and MCH (*p* = 0.000) values varied across cycles, whereas MCV values were significantly higher (*p* = 0.0466) in the shaded ponds. Overall, two-factor ANOVA analyzing the two independent factors (cycle and shading setup) against the hematological profiles revealed significant variations (*p* < 0.05) (Figure 3A–E). The values of WBCs, PCt, MCV, MCH, and MCHC were not affected by the interaction, and substantial results were only observed when the cycle and shading setup were analyzed separately.

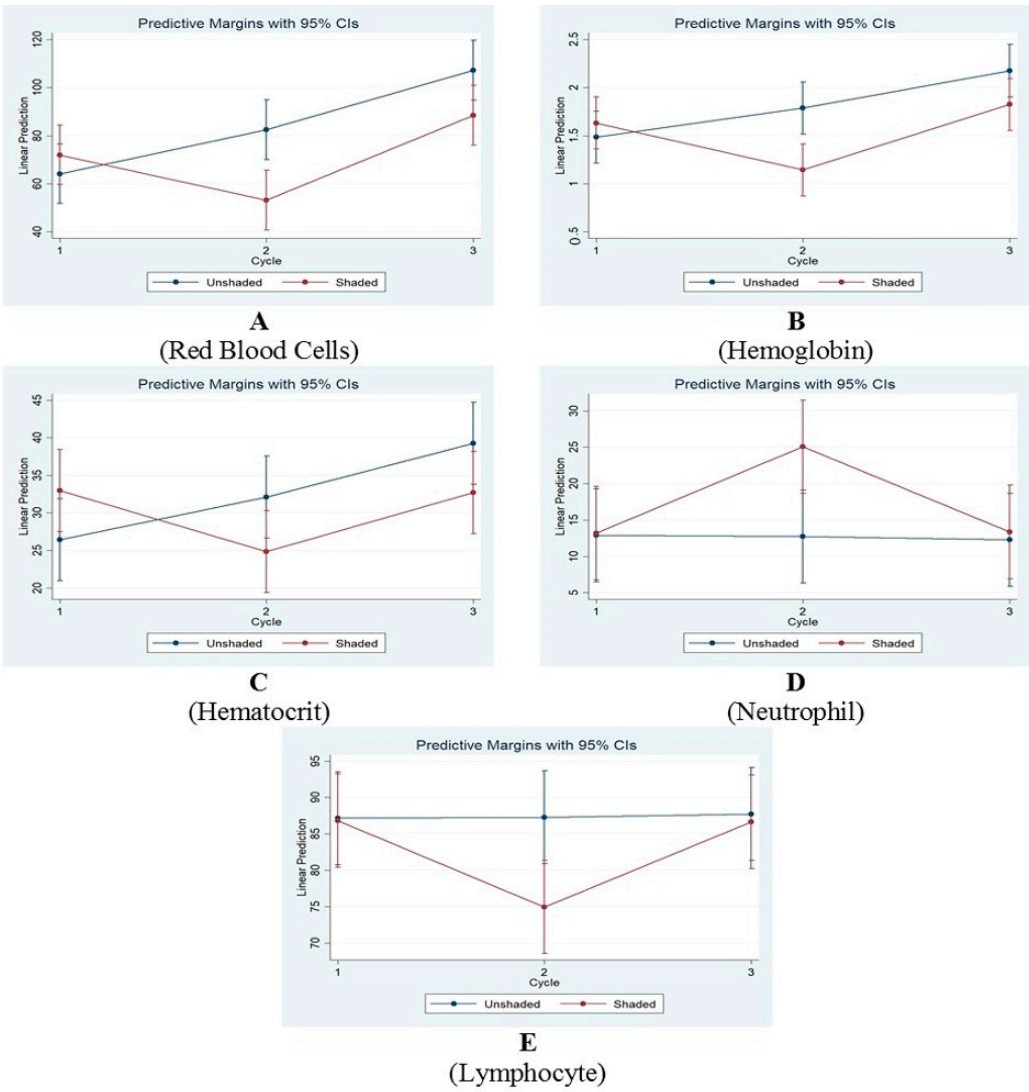

**Figure 3.** Two-factor ANOVA analyzing the two independent factors (cycle and shading setup) against the hematological profiles, including red blood cell (**A**), hemoglobin (**B**), hematocrit (**C**), neutrophil (**D**), and lymphocyte (**E**) parameters.

### 3.4. Biochemical Study

The biochemical profile was calculated and tabulated in the control and treated ponds (Tables 6 and 7). Two-factor ANOVA analyzing the two independent factors (cycle and shading setup) against the biochemical profile revealed significant variations ($p < 0.05$) (Figure 4A–E). Chole, TP, and Alb values were significantly affected by the interaction of cycle and shading setup at $p = 0.0060$, $p = 0.0065$, and $p = 0.00550$, respectively, while Glu, AST, ALT, ALP, and $Na^+$ values were not. Some other parameters showed significant differences and were only affected by such factors independently. For instance, cortisol, Glb, $K^+$, $Cl^-$, $Mg^{2+}$, and TCa values showed significant differences and were only affected by the cycle at $p = 0.0221$, $p = 0.0000$, $p = 0.0042$, $p = 0.025$, $p = 0.0010$, and $p = 0.0006$, respectively. Similarly, AST and ALT values showed significant differences and were affected by both cycles at $p = 0.0000$ and shading setup at $p = 0.0000$ and $p = 0.0040$, respectively.

Regarding the mean values of biochemical parameters (Table 7), only AST and ALT concentrations were significantly higher ($p < 0.05$) in the shaded pond, while all other parameters (i.e., Cor, Glu, Chole, TP, Alb, Glb, ALP, $K^+$, $Na^+$, $Cl^-$, $Mg^{2+}$, and TCa) were comparable. The values of Glu, Alb, Mg, AST, and ALT were relatively higher in the shaded ponds, whereas those of Cor, Chole, TP, ALP, K, Na, Cl, and TCa were found to be lower.

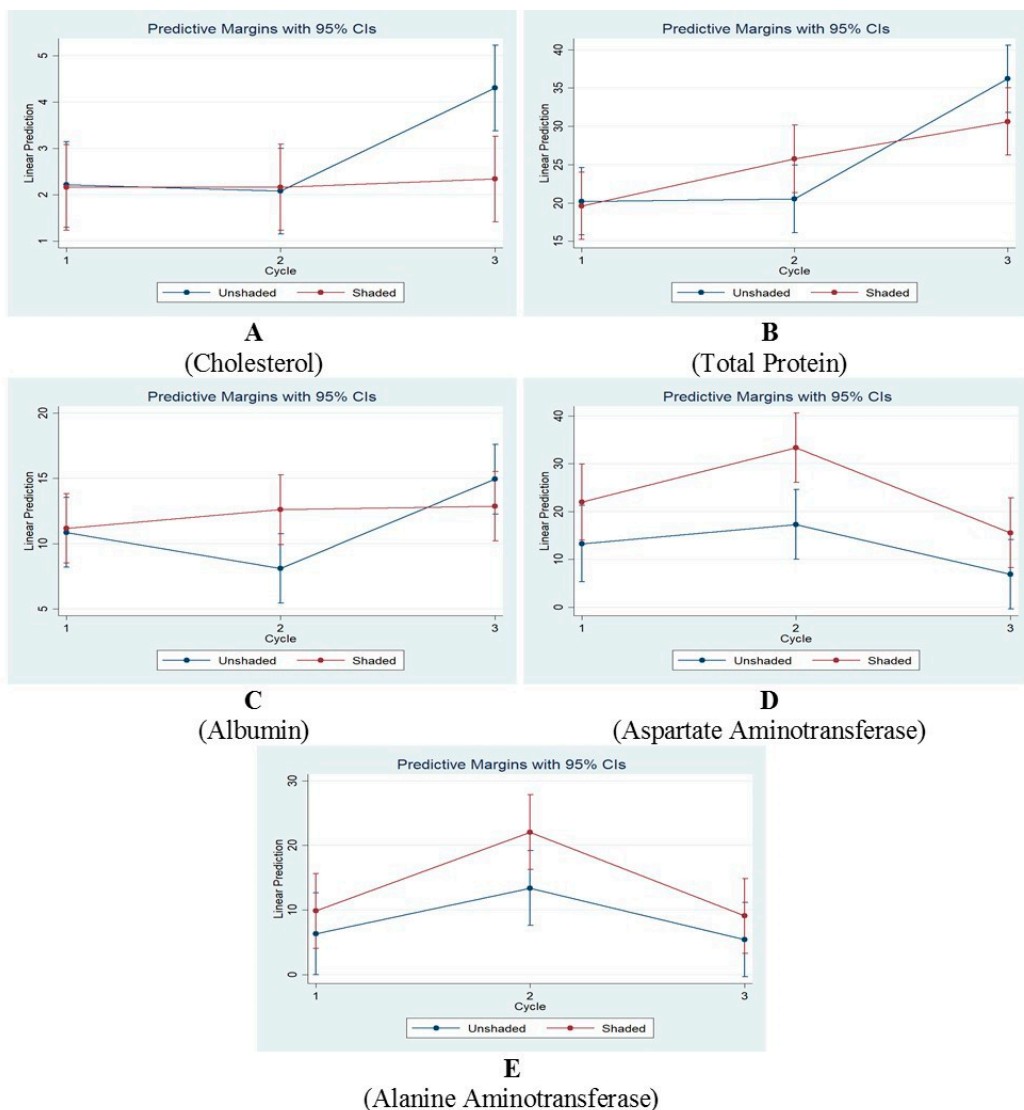

**Figure 4.** Two-factor ANOVA analyzing the two independent factors (cycle and shading setup) against the biochemical parameters, including cholesterol (**A**), total protein (**B**), albumin (**C**), aspartate aminotransferase (**D**), and alanine aminotransferase (**E**).

### 3.5. Canonical Correlation Analysis (CCA)

The CCA tables (CCA1, CCA2, and CCA3) were used to interpret the relationships between the water quality, hematological, and biochemical parameters. Both WT and DO were positively correlated with each other (Table 8). The correlation matrix of hematological and biochemical parameters is presented in Tables 9 and 10, respectively. As illustrated, there was a strong positive correlation between WBC and Hct, Hgb and Hct, RBC, Hgb and Hct, Na$^+$, and Alb. Moreover, both Glu and ALT were positively correlated to electrolyte K$^+$, while cortisol was negatively correlated to Alb.

**Table 8.** Correlation matrix of water quality parameters.

| | DO | | WT | |
|---|---|---|---|---|
| | **Morning** | **Afternoon** | **Morning** | **Afternoon** |
| AM_DO | 1.0000 | | | |
| PM_DO | 0.9299 * | 1.0000 | | |
| AM_WT | 0.4539 | 0.7402 * | 1.0000 | |
| PM_WT | 0.6124 * | 0.8551 * | 0.9706 * | 1.0000 |

* Significance at $p \leq 0.05$. DO, dissolved oxygen; WT, water temperature.

**Table 9.** Correlation matrix of hematological parameters.

|       | WBCs | RBCs | HGB | HCT |
|-------|------|------|-----|-----|
| WBCs | 1.0000 | | | |
| RBCs | 0.3710 | 1.0000 | | |
| Hgb | 0.2931 | 0.9648 * | 1.0000 | |
| Hct | 0.5496 * | 0.8459 * | 0.7392 * | 1.0000 |

* Significance at $p < 0.05$.

**Table 10.** Correlation matrix of biochemical parameters.

|      | Cortisol | Glu | K | Na | Alb | Alt |
|------|----------|-----|---|-----|-----|-----|
| Cor | 1.0000 | | | | | |
| Glu | −0.1447 | 1.0000 | | | | |
| $K^+$ | 0.1147 | 0.1789 | 1.0000 | | | |
| $Na^+$ | −0.3888 | 0.8811 * | 0.0536 | 1.0000 | | |
| Alb | −0.6378 * | 0.2638 | −0.2220 | 0.5551 * | 1.0000 | |
| ALT | −0.2072 | −0.1759 | 0.5361 * | −0.0374 | 0.0417 | 1.0000 |

* Significance at $p < 0.05$.

Regarding the correlation between water quality and hematological parameters, the best linear combination between them was obtained at 0.96 (Table 11). WBC and Hgb were the main blood indices in CC1 that were correlated to all water parameters. In CC2, RBC and Hgb were the parameters greatly affecting the DO and PM WT. In CC3, PM DO and WT in AM or PM were the major coefficients affecting water quality. Likewise, both water quality and biochemical parameters were closely correlated at 0.97 (Table 12). Cortisol, Glu, and $K^+$ were the major biochemical parameters in CC1 that were correlated to all water parameters, the same as with CC2. In CC3, Alb was the major coefficient affecting the AM WT. Lastly, in CC4, the DO in both AM and PM was the factor contributing to water quality parameters. Moreover, the correlation between hematological parameters and biochemical parameters was found to be statistically significant at 0.96 (Table 13). Cortisol, $K^+$, and ALT were the major biochemical parameters in CC1 that were correlated to WBC, RBC, and Hgb. In CC2, cortisol and $K^+$ were the main contributors affecting hematological parameters.

**Table 11.** Canonical correlation analyses between hematological profiles and water quality parameters.

|   | CC1 | CC2 | CC3 | CC4 |
|---|-----|-----|-----|-----|
| Canonical correlation coefficient | 0.9639 ** | 0.9000 ** | 0.8578 ** | 0.3165 |
| Standardized canonical coefficients for the hematological parameters | | | | |
| WBCs | 0.6572 * | −0.0123 | −0.7105 | 1.2837 |
| RBCs | −0.5346 | 3.8764 * | 1.0083 | 2.0095 |
| Hgb | 1.3193 * | −3.6007 * | 0.0155 | −1.2506 |
| Hct | −0.2418 | −0.0547 | −0.2952 | −2.1678 |
| Standardized canonical coefficients for the water quality parameters | | | | |
| AM_DO | −7.7032 * | 5.4862 * | 2.4225 | −1.8528 |
| PM_DO | 10.1404 * | −8.1625 * | −5.9993 * | 3.3276 |
| AM_WT | −1.8084 * | 1.7332 | −4.8970 * | −2.8808 |
| PM_WT | −1.7801 * | 2.7274 * | 8.2713 * | 1.5154 |

* Most important coefficients; ** significance at $p < 0.05$.

**Table 12.** Canonical correlation analyses between biochemical profiles and water quality parameters.

| | CC1 | CC2 | CC3 | CC4 |
|---|---|---|---|---|
| Canonical Correlation Coefficient | 0.9698 ** | 0.9675 ** | 0.8070 ** | 0.6085 |
| Standardized canonical coefficients for the biochemical parameters | | | | |
| Cor | 0.6819 * | −0.7661 * | 0.6282 | 0.5375 |
| Glu | 0.5817 * | 0.7888 * | 0.1783 | 1.1354 |
| $K^+$ | 0.3396 * | 0.5162 * | 0.2286 | −0.6224 |
| $Na^+$ | −0.3353 | −0.3648 | −0.5744 | −0.3751 |
| Alb | −0.1065 | −0.1469 | 1.2720 * | 0.5154 |
| ALT | 0.1710 | −0.0067 | 0.5244 | 0.1719 |
| Standardized canonical coefficients for the water quality parameters | | | | |
| AM_DO | 2.5705 * | 3.0767 * | −2.3101 | −8.7937 * |
| PM_DO | −5.8858 * | −3.3630 * | 2.0054 | 12.9052 * |
| AM_WT | −3.9112 * | −1.3316 * | −3.8975 * | −2.5087 |
| PM_WT | 7.7767 * | 2.3156 * | 2.7127 | −2.8495 |

* Most important coefficients; ** significance at $p < 0.05$.

**Table 13.** Canonical correlation analyses between hematological parameters and biochemical profiles.

| | CC1 | CC2 | CC3 | CC4 | CC5 |
|---|---|---|---|---|---|
| Canonical correlation coefficient | 0.9613 ** | 0.9027 ** | 0.7551 | 0.6140 | 0.2428 |
| Standardized canonical coefficients for the biochemical parameters | | | | | |
| Cor | 0.6467 * | −0.9950 * | 0.3943 | 0.5198 | 0.1168 |
| Glu | −0.4169 | −0.0598 | −1.1192 | 0.7670 | −0.5938 |
| K+ | 0.5407 * | 0.4905 * | 0.8814 * | −0.6632 | −0.5638 |
| Na | 0.4636 | −0.2759 | 1.0263 | −1.2460 | 1.6634 |
| Alb | −0.1743 | −0.2345 | 0.8952 | 0.9808 | −0.6221 |
| ALT | 0.2925 * | 0.1291 | −0.6186 | 0.9984 | 0.6875 |
| Standardized canonical coefficients for the hematological parameters | | | | | |
| WBCs | −0.5574 * | 0.2547 | 0.1556 | 0.1665 | −2.0119 |
| RBCs | 2.6496 * | −1.3751 | 0.5989 | −5.7538 * | 1.2531 |
| Hgb | −1.5591 * | 0.0257 | 0.7708 | 4.5073 * | −0.6533 |
| Hct | −0.3965 | 0.1456 | −1.7538 * | 1.5066 | 0.7254 |

* Most important coefficients; ** significance at $p < 0.05$.

## 4. Discussion

To understand the potential effects of net shading on tilapia breeders, the water parameters as well as the biochemical and hematological profiles of Nile tilapia breeders were investigated in the current study. The findings of this study revealed that the mean values of WT were significantly decreased by shading, which is particularly relevant in consideration of temperature warming. In aquaculture conditions, seasonal changes in water temperature have an impact on the immune system as well as other physiological processes such as growth efficiency and reproduction. However, abrupt variations in water temperature can cause stress, affecting the physiologic activities of the fish [33]. In theory, fish in warmer waters consume more energy as metabolic energy demands double with every 10 °C rise in body temperature [34]. In cases where fish are subject to thermal fluctuations, there may be an unnecessary expenditure of high energy, which would contribute to the physical growth of the fish [35]. Shading decreased water temperature and cool water may hold more oxygen than warm water [36], and the overall difference in DO concentrations between the two setups was not significant. However, during Cycle 1, it was observed that there was significant difference between the two setups. This has been corroborated by the study of Memis et al. [37], wherein no significant difference in the DO concentration was observed during the first year, except towards the end of

summer periods wherein the potential benefits of shade were reduced since the presence of phytoplankton was affected by the light intensity.

Surprisingly, DO concentrations were affected by diurnal variation. During the day, photosynthesis increases oxygen levels in the water. However, breathing reduces dissolved oxygen, resulting in a lower dissolved oxygen concentration in the early morning [38]. Furthermore, the rates of both photosynthesis and respiration increase with increasing temperature, which is why both severe hypoxia and hypoxia are common in summer and less frequently encountered in cooler months [39]. This was similar to the results seen in the study wherein morning and afternoon DO concentrations were far apart. Herein, the mean DO values were slightly lower and below the minimum threshold levels for survival (2 mg/L) in the unshaded ponds compared to the shaded ones. It is also possible for low oxygen availability to decrease assimilation. Hence, growth and reproduction will be negatively affected in conditions of high WT and low DO concentration. This may explain why the condition factor of fish in the shaded pond, though not statistically significant, was higher than in the unshaded pond. Further, in the study of Ribeiro et al. [40], fish swim much slower under shade than out of the shade, consuming less energy and conserving more nutrients. The condition factor (CF) of <1.0 indicates that the fish is in poor health condition while CF > 1.4 shows that the fish is in excellent condition [28]. The results showed that CF was always >1.4, so the setup does not compromise the health status of the fish. The current study also corroborates the CF result in the study of Vivanco-Aranda et al. [41], wherein fish CF slowed down during summer compared to the spring with no statistical difference. The similarity could be attributed to the condition of the water temperature.

Blood parameters depend greatly on seasonal variation and may be influenced by changes in water temperature and dissolved oxygen [28,42]. Regarding the hematological analysis, the RBC count, Hgb concentration, and HCT percentage were lower in the shaded ponds than in the non-shaded counterparts, with RBC count and Hgb being statistically significant, confirming the results of the study by Panase and colleagues [43] in which the RBC count, Hgb concentration, and HCT were decreased as the temperature decreased. Moreover, the result showing less hematological disturbance proves the importance of the pre-acclimation phase improving fish's tolerance to change in environmental condition [44]. Higher temperature and lower oxygen concentration are consistent with the unshaded setup of the present study. These results in higher erythrocytes required to carry oxygen around the fish's body in warm water, as the oxygen is less readily available. Under these conditions, values of blood parameters such as HCT, RBC, HGB, and MCHC may increase [45], in agreement with the results of the present study.

Herein, the increased hematocrit levels may also be a result of higher exposure to stress; however, this is often associated with increased cardiac output from intense exercise compared to hypoxia [46]. Sudden fluctuations in temperature may also affect the immune response of fish. In a study by Ndong and colleagues [47], sudden temperature changes (from 27 °C to either 19–23 °C or 31–35 °C) reduced the pathogen resistance and immune response of *Oreochromis mossambicus*. The decreased immune function may be an indirect result of the neuroendocrine response, as immunosuppressive cortisol is released during thermal stress [17,48]. Moreover, several pathogens thrive better at elevated temperatures [49], compounding the risk of immunocompromised fish to infectious diseases [17]. WBC count is lower in a shaded pond. A similar result was found in mackerels, which gives lower results at low temperature [41]. Increased WBC counts throughout the summer and higher temperatures have been linked to infection caused by a number of diseases, indicating a fish protective response [41,45]. Nevertheless, in this study, no sign of disease was observed.

As mentioned earlier, cortisol is released during thermal stress. Although there was no significant difference in cortisol levels, cortisol levels were higher in fish in the unshaded pond than in the shaded one, which may indicate lower stress in fish in shaded ponds because of lower water temperature. It has been reported that plasma cortisol level changes depending on the stressor and the level of stress imposed on the fish including multiple

stresses [50–52]. In our study, the average cortisol level in the unshaded setup was at $244.2 \pm 81.0$ nmol/L compared to the shaded setup at $239.4 \pm 113.1$ nmol/L. This clearly showed that the fish population in the study under a non-shaded setup was under stress. Despite that fact, it was also observed that shading decreases cortisol level and therefore decreases the stress level on the fish population.

Moreover, the increase in water temperature could enhance carbohydrate utilization and liver glycolytic, gluconeogenic, and lipogenic capabilities in cultured fish [53,54]. The study of Fazio et al. [28] reported an increase in blood glucose in cold temperature. Higher temperatures would promote glucose absorption into cells [55]. This supports the result of the present study, which recorded an increased level of blood glucose under the shaded setup. The highest blood glucose level was observed in the shaded pond compared to the unshaded one and thus was correlated to the lowering of the temperature. Moreover, the average glucose concentration between the shaded and the unshaded setup was not significantly varied. Similarly, the lower levels of sodium and potassium observed in the shaded pond fish may be related to lower energy intake from increased dissolved oxygen and thus reduced hematopoiesis [32]. Further, during exposure to acute stressors, spikes in adrenaline cause vasoconstriction and increased cardiac output, increasing gill permeability. This results in an increase in ion transfer that alters specific ion concentrations in the plasma [23]. Despite this, the levels of sodium and potassium were still within the reference ranges published by Mauel and colleagues [56]. Although higher levels of cholesterol were seen in fishes in the unshaded pond, this may provide information on mobilized energy stores, but cholesterol responses to stressors are often inconsistent [23].

TP concentrations were decreased under the shading setup (Table 7), and AST and ALT posed significantly higher concentrations than the other parameters in the shaded pond (Table 6). Increased temperature can cause structural liver alterations, thereby reducing aminotransferase activity, decreasing deamination capacity, and impairing the control of fluid balance [57]. In this study, the use of shading decreased TP, thus increasing aminotransferase activity, increasing deamination capacity and balancing the electrolytes without greatly affecting the fish. These results are consistent with the study of Jiang et al. [20], which revealed that total protein and albumin levels increased at 21 °C, elucidating that high temperatures have caused tissue damage as protein aids in tissue repair. This highlights that high concentrations of oxygen may reduce the extent of tissue damage and boost its recovery [19]. Moreover, ALT and AST play important roles in protein metabolism and are found in liver and heart muscle cells, respectively, in fish [58]. Significant increase and variations in AST and ALT concentration are predominantly affected by cycles and not solely by shading setup. During Cycle 2, DO concentration was significantly decreased, causing hepatic damage, which in turn released enzymes into the bloodstream. In the present study, AST and ALT at different WT and DO changed, but the values remained in the normal range, indicating that the fish were in good condition [59].

While there was a considerable variation in AST and ALT concentrations between the two setups, the other parameters yielded comparable results. Moreover, $Mg^{2+}$ had a higher level in the shaded pond while $Ca^{2+}$ had lower concentrations. This is in contrast with the study of Fazio et al. [28], where both have a similar trend observed where lower concentrations were observed in autumn, which indicates lower temperatures. Magnesium acts as an enzyme cofactor and is essential for the structural component of cell membranes in organs and muscle tissues, while calcium aids in muscular contraction and neurological processes. These levels serve at par with the above findings on protein, proposing that increased activity is indicative of tissue damage, which is exhibited at higher temperatures. Meanwhile, the lower concentrations of K, Na, and Cl in the tilapia situated in shaded ponds may be explained by the levels of cortisol found in this study, as well [60]. It must be noted that cortisol induces the stimulation of the Na-K ATPase in the plasma membrane and in the regulation of plasma ionic composition. Accordingly, our findings are consistent with the results of Musa et al. [13], which revealed a positive trend: as the cortisol levels in fish increased at higher temperatures, $K^+$, $Na^+$, and $Ca^{2+}$ were also higher, indicating that

temperature influences the plasma ion concentrations. A similar trend was also observed in the study of Phinrub et al. [61], where the electrolyte concentrations increased at higher temperatures, particularly at 31 °C–37 °C. These findings may also be attributed to the fact that changes in water temperature can induce an impaired osmoregulation ability [62]. Potassium is of significant interest as well, as it has been shown to exhibit an important role in the sodium–potassium pump. However, further studies are necessary to consider salinity as a factor that would potentially affect the ion concentrations of *O. niloticus*, focusing on plasma osmolality.

## 5. Conclusions

This study demonstrated that the water quality, hematological, and biochemical parameters collected from different pond shading setups, cycles, and groupings were all related to each other. There was a positive correlation between hematological parameters and water quality parameters (r = 0.96). Further, there was a correlation between hematological and biochemical parameters (r = 0.96) and a correlation between biochemical parameters and water quality parameters (r = 0.97). The hematological and biochemical changes observed in this study suggest a possible association with stress due to an increase in temperature. In conclusion, the hematological and biochemical changes observed in this study gave insight into the potential use of shading in freshwater ponds to reduce fish stress during high temperatures. This study also provides evidence for mitigation strategies to address temperature warming due to climate change that may negatively affect the survival and reproduction of Nile tilapia in aquaculture sectors, especially in tropical and subtropical countries.

**Author Contributions:** Conceptualization, G.B.D., E.M.V.C., C.R., M.M., P.P. and M.D.S.; methodology, G.B.D., E.M.V.C., C.R., M.M., P.P. and M.D.S.; data collection, G.B.D., E.M.V.C., C.R., M.M., P.P. and M.D.S.; formal analysis, all authors; writing of the manuscript, all authors; supervision, G.B.D.; funding acquisition, G.B.D. All authors have read and agreed to the published version of the manuscript.

**Funding:** This work was funded by the International Foundation for Science (J-2-A-6028-1).

**Institutional Review Board Statement:** This study was conducted according to the national and institutional guidelines for the protection of animal welfare of CLSU and approved by the Animal Ethics Review Committee of Suez Canal University (AERC-SCU), Egypt (2023006, 28 December 2022).

**Data Availability Statement:** The data that support the findings of this study are available from the corresponding author upon request.

**Acknowledgments:** The authors would like to acknowledge the following academic and research institutions: the University of Santo Tomas—The Graduate School, Central Luzon State University—Freshwater Aquaculture Center, and the International Foundation for Science.

**Conflicts of Interest:** The authors declare no conflict of interest.

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
