# Peer review of "Potential Influence of Shading in Freshwater Ponds on the Water Quality Parameters and the Hematological and Biochemical Profiles of Nile tilapia (Oreochromis niloticus Linnaeus, 1758)"

_fishes, doi:10.3390/fishes8060322_

Round 1

Reviewer 1 Report (Previous Reviewer 1)

Authors have responed my concerns, and I have no comment.

Reviewer 2 Report (Previous Reviewer 3)

Dear Authors, 

Thank you for the additional clarification, as a result, the manuscript has been significantly improved. The manuscript now meets the standards of the journal. 

Best wishes

The writing as well as the quality of the language have been improved. 

This manuscript is a resubmission of an earlier submission. The following is a list of the peer review reports and author responses from that submission.

Round 1

Reviewer 1 Report

This study explored the effects of net shading on the water quality of the culture pond and the serum biochemical profiles of Nile tilapia. The results showed that the average temperature but not dissolved oxygen in the pond with net shading was decreased compared to the unshaded pond. The results of the biochemical data indicate that red blood cells, hemoglobin, and lymphocyte counts were significantly lower in fish raised in shaded ponds. In addition, cortisol, Glb, K+, Cl-, Mg2+, TCa values showed significant differences and were only affected by the cycle. The studies provide some reference information for the culture of Nile tipapia. I have some comments as follows:

1. The present introduction is too brief. The authors should extend the introduction let readers to understand why the biochemical parameters measured in the article are necessary. Besides, the authors did a lot of biochemical analysis but did not discuss these data well or even not mentioned these data in the discussion. Authors also have to rewrite and discuss these biochemical data suitably.  

2. The authors should provide the animal care approval in the text.

3. Please provide the information of feeding condition to the cultured fish.

4. Figure 3 and 4: Please provide the complete figure legend.

5. All tables: please provide the N values. In addition, please unify the format of tables. Some places showed the mean+SD but absent in some tables.  

6. The authors should identify the full name of the abbreviation at the place where the term is first mentioned in each section. Or the authors can create a section for the abbreviation. 8. Line 4: “Oreochromis niloticus” should be italic.

7. Lines 140-143: please describe which figure or table you mention.

8. Lines 32, 114, 115 and other places: K+, Cl-, and other ions….please superscript the “+ “ and “-“.

Reviewer 2 Report

Potential influence of shading in freshwater ponds on the water quality parameters, hematological and biochemical profiles of Nile tilapia (Oreochromis niloticus Linnaeus, 1758)

Overall a good study. Acceptable with minor modifications

Introduction

Use new references

2. Materials and Methods

2.4. Fish Biometric Index

Provide other indicators,

Viscerosomatic index (%), VSI =100 × visceral weight (g)/final fish weight (g),

Hepatosomatic index (%), HIS =100 × liver weight (g)/final fish weight (g),

 Carcass yield (%), CY = 100 × total fish weight – visceral weight (g)/total fish weight (g).

Use new references in the discussion section

Reviewer 3 Report

The manuscript submitted by Dayrit and colleagues investigate the effects of shaders (i.e., protecting the water from direct sunlight) on the water quality (e.g., temperature and dissolved oxygen) and other haematological and biochemical parameters in Nile tilapia. Although there are interesting results executed and may be attractive to tilapia farmers, the manuscript is incomprehensible in several ways.

The authors claimed that by using a net shader over the tilapia farm, they reduced the consequences of global warming as per stated in the abstract and concluding statement in the discussion. I understand the point you are trying to make here at large, however, in the context of this research, there is very unlikely a relation between global warming and the effectiveness of shaders in preventing its consequences in tilapia ponds. The concept of global warming must be removed from the entire manuscript. Moreover, this research lacks controls, as it is difficult to determine if the changes in haematological/biochemical changes between groups were simply because of shaders or if there are natural physiological occurrences as the fingerlings grow. As stated in table 1, the water temperature differences are negligible between shaded and unshaded ponds, and as this kay factor isn’t changed, I am not convinced that all other parameters tested in this study are correlated to the effect of the shaders. Also, the in-flow water supply to each post was not stated if there is one for each pond the temperature data is not valid. In general, the MS requires a better hypothesis in the introduction and clearly states how this research addresses the hypothesis question (s).

Material and methods must be improved by adding more details for methods used or appropriate citations if the method was previously used by others. There are also no replicates presented. It is not clear if the data is statistically homogenised. In general, results are poorly presented, for instance, figure 4 needs better graph presentation and legend. The discussion is pre-mature and must improve with better physiological interpretations and better connections with previous similar studies. Discussion as of now is a mixture of repetitive results carried away, which currently consist of assumptions with no evidence from this study (e.g., lines 243-244; 264-271; 278-281; 283-285; 286-291, etc..). Also, in the discussion, some results were incorrectly interpreted which requires the authors’ attention and revision prior to submission.  

Based on the concerns regarding the interpretation and presentation of the results and following the assessment of these discordant, I have concluded that your manuscript is not suited for publication in the journal of Fishes/MDPI at this time, but there are other journals that may better suit your work.